# Comparative genomics reveals the origin of fungal hyphae and multicellularity

Enikő Kiss[1,2], Botond Hegedüs[1], Máté Virágh[1], Torda Varga [1,2], Zsolt Merényi[1], Tamás Kószó[1], Balázs Bálint[1], Arun N. Prasanna [1,6], Krisztina Krizsán[1], Sándor Kocsubé[3], Meritxell Riquelme [4], Norio Takeshita [5] & László G. Nagy[1]

Hyphae represent a hallmark structure of multicellular fungi. The evolutionary origins of hyphae and of the underlying genes are, however, hardly known. By systematically analyzing 72 complete genomes, we here show that hyphae evolved early in fungal evolution probably via diverse genetic changes, including co-option and exaptation of ancient eukaryotic (e.g. phagocytosis-related) genes, the origin of new gene families, gene duplications and alterations of gene structure, among others. Contrary to most multicellular lineages, the origin of filamentous fungi did not correlate with expansions of kinases, receptors or adhesive proteins. Co-option was probably the dominant mechanism for recruiting genes for hypha morphogenesis, while gene duplication was apparently less prevalent, except in transcriptional regulators and cell wall - related genes. We identified 414 novel gene families that show correlated evolution with hyphae and that may have contributed to its evolution. Our results suggest that hyphae represent a unique multicellular organization that evolved by limited fungal-specific innovations and gene duplication but pervasive co-option and modification of ancient eukaryotic functions.

[1] Synthetic and Systems Biology Unit, Institute of Biochemistry, BRC-HAS, Temesvari krt 62, 6726 Szeged, Hungary. [2] University of Szeged, Faculty of Science and Informatics, Aradi vertanuk tere 1., 6720 Szeged, Hungary. [3] Department of Microbiology, University of Szeged, Faculty of Science and Informatics, Kozep fasor 52, 6726 Szeged, Hungary. [4] Department of Microbiology, Centro de Investigación Científica y de Educación Superior de Ensenada, Carr Tijuana-Ensenada 3918, C.I.C.E.S.E, 22860 Ensenada, Baja California, Mexico. [5] Microbiology Research Center for Sustainability (MiCS), Faculty of Life and Environmental Sciences, University of Tsukuba, 1 Chome-1-1 Tennodai, 305-8572 Tsukuba, Japan. [6]Present address: Red Sea Science and Engineering Research Center, 4700 King Abdullah University of Science and Technology (KAUST), Thuwal 23955-6900, Saudi Arabia. Correspondence and requests for materials should be addressed to L.G.N. (email: lnagy@fungenomelab.com)

The evolution of multicellularity (MC) is considered one of the major transitions in the history of life[1]. Multiple bacterial and eukaryotic lineages underwent this major transition[2–7], in each case arriving at a unique solution to the challenges of multicellular organization[6]. Among eukaryotes, multicellularity appears to have arisen via either clonal and aggregative mechanisms[5,6,8,9], which differ in how multi-celled precursors adhere, cooperate, communicate and functionally diversify[3,10,11].

Fungi constitute one of the three kingdoms where a majority of extant species are multicellular[12], yet, the origins of fungal multicellularity remain obscure. While most multicellular lineages can be recognized as being either clonal or aggregative by comparisons to their unicellular relatives, fungal multicellularity has been recalcitrant to such categorization[6,13]. Multicellularity in fungi refers to a thallus made up of hyphae, thin, tubular structures that grow by apical extension to form a mycelium that explores and invades the substrate. Hyphal multicellularity exhibits several unique properties that distinguish it from clonal and aggregative multicellularity, raising the possibility that its evolution may follow markedly different principles[7].

First, hyphae might have evolved by the gradual elongation of substrate-anchoring rhizoids of early fungi[14–16], through multinucleate intermediates, in contrast to clonal and aggregative lineages, where the first multi-celled clusters probably emerged via related cells sticking together (e.g., choanoflagellates[17]), or gathering to form a syncytial body (e.g., ichthyosporeans)[18]. Because early hyphae were uncompartmentalized, their evolution could have bypassed the need to resolve group conflicts and align the fitness of individual cells[7]. Alternatively, it is possible that conflicts are resolved at the level of individual nuclei[19]. Second, hyphae maximize foraging and nutrient assimilation efficiency and minimize competition for nutrients by a fractal-like growth mode[20–22]. The mechanism of the origin of hyphae differs from that of other multicellular lineages where selection for increased size possibly helped avoiding predation[2]. Hyphae might have also facilitated the transition of fungi to terrestrial life, by bridging nutrient-rich and nutrient-poor habitats[23] and confer immense medical relevance to pathogenic species[24]. Hyphae of extant fungi rarely stick to each other in vegetative mycelia and adhesion becomes key only in fruiting bodies[25,26]—which, in terms of complexity level, resemble multicellular metazoans and plants[7,27] —or in the attachment to host surfaces[28]. Thus, whereas in most multicellular lineages adhesion, cell–cell cooperation, communication and differentiation represent the main hurdles to the emergence of multicellular precursors[3,6,29,30], fungi might have had different obstacles to overcome.

While the evolutionary origins of hyphae are obscure, information on the molecular and cellular basis of hypha morphogenesis is extensive (for recent reviews see refs. [31–34]), permitting evolutionary genomic analyses. Hypha morphogenesis builds on cell polarization networks[35], the exo- and endocytotic machinery[36], long-range vesicle transport as well as fungal-specific traits such as cell wall synthesis and assembly[37], and the selection of branching points and septation sites[38]. A key structure of hyphal growth is the Spitzenkörper[39], which acts as a distribution center for vesicles transporting cell wall materials and other factors to the hyphal tip. The cytoplasmic microtubule network provides the connection between vesicle cargo from the ER and Golgi and the Spitzenkörper, from where vesicles move to the hyphal tip and secrete their content for building the cell wall and provide surface expansion. Further key processes include the recycling of excess membrane in the subapical zone, the activation of cAMP pathways and mitogen-activated protein kinase (MAPK) cascades and finally the transcriptional control of morphogenesis (reviewed in refs. [34,40–42]).

A complex hyphal thallus has been reported from a 407 million-year-old fossil Blastocladiomycota[43], whereas Glomeromycotina-like hyphae and spores were preserved 460 million years ago[44,45] indicating that hyphal growth dates back to at least the Ordovician. Most Dikarya and Mucoromycota grow true hyphae, whereas a significant diversity of forms exists in the Blastocladiomycota, Chytridiomycota and to a smaller extent the Zoopagomycota. The Chytridiomycota is dominated by unicellular forms that anchor themselves to the substrate by branched, root-like rhizoids[22,45]. These structures have been hypothesized as the precursors to hyphae[14,46]. An alternative hypothesis designates hypha-like connections in the thalli of polycentric chytrid fungi (e.g. *Physocladia*) as intermediates to true hyphae[15]. Like chytrids, most Blastocladiomycota form mono- or polycentric, unicellular thalli, although some species form wide, apically growing structures resembling true hyphae (e.g., *Allomyces*) or narrow exit tubes on zoosporangia (e.g., *Catenaria* spp.)[22,45,47]. In spite of these intermediate forms, the prevalence of unicellular forms in these phyla indicates their unicellular ancestry and suggests potential convergent origins of hypha-like structures[15].

Here we examine how the genetic toolkit of hyphal multicellularity was assembled during evolution by reconstructing the evolutionary history of known hypha morphogenesis genes and by systematically searching fungal genomes for gene families whose evolution correlates with that of hyphae. We analyze the genomes of 4 plesiomorphically unicellular, 41 hyphal and 13 secondarily simplified (yeast-like) fungi as well as 14 non-fungal relatives. We identify a multitude of small changes in hyphal-morphogenesis gene families that correlate with the evolution of hyphae, including co-option and exaptation of ancient eukaryotic genes, limited gene family diversification and alterations of gene structure. Correlated patterns of gene duplication and loss that correlate with the origin of hyphal multicellularity were detected for 414 gene families, providing further candidate key genes. These data indicate that many small changes rather than one major innovation, underlie this key fungal innovation, compatible with evolutionary tinkering[48].

## Results

**Hyphae evolved in early fungal ancestors.** To understand the origin of hyphae, we constructed a species phylogeny representing 72 species (Supplementary Data 1) by maximum likelihood and Bayesian MCMC analysis of a supermatrix of 455 single-copy orthologs (75,224 characters, Fig. 1a, Supplementary Fig. 1).

Our species phylogeny is strongly supported and recapitulates recent genome-based phylogenies of fungi[49–52], with the Rozellomycota, Blastocladiomycota and the Chytridiomycota splitting first, second and third off of the backbone, respectively (ML bootstrap: 100%). We next scored species for their ability to form multicellular hyphae (Fig. 1a, Supplementary Data 1) and performed ancestral character state reconstructions using Bayesian MCMC. This suggested that hyphae evolved from unicellular precursors in some of the earliest fungal ancestors. The distribution of posterior probability values indicated three nodes as the most likely origins of hyphal multicellularity, which represent the split of Blastocladiomycota, Chytridiomycota and Zoopagomycota, and are referred hereafter to as BCZ nodes (Fig. 1a). The posterior probability for the hyphal state started to rise in the most recent common ancestor (MRCA) of the Blastocladiomycota and higher fungi (PP: 0.53, Fig. 1a) and increased to 0.68 and 0.92 in the next two nodes up in the tree. This suggests that hyphae evolved either in one of the BCZ nodes or that its evolution was a gradual process unfolding in these three nodes. This distribution also reflects the diverse hypha-like

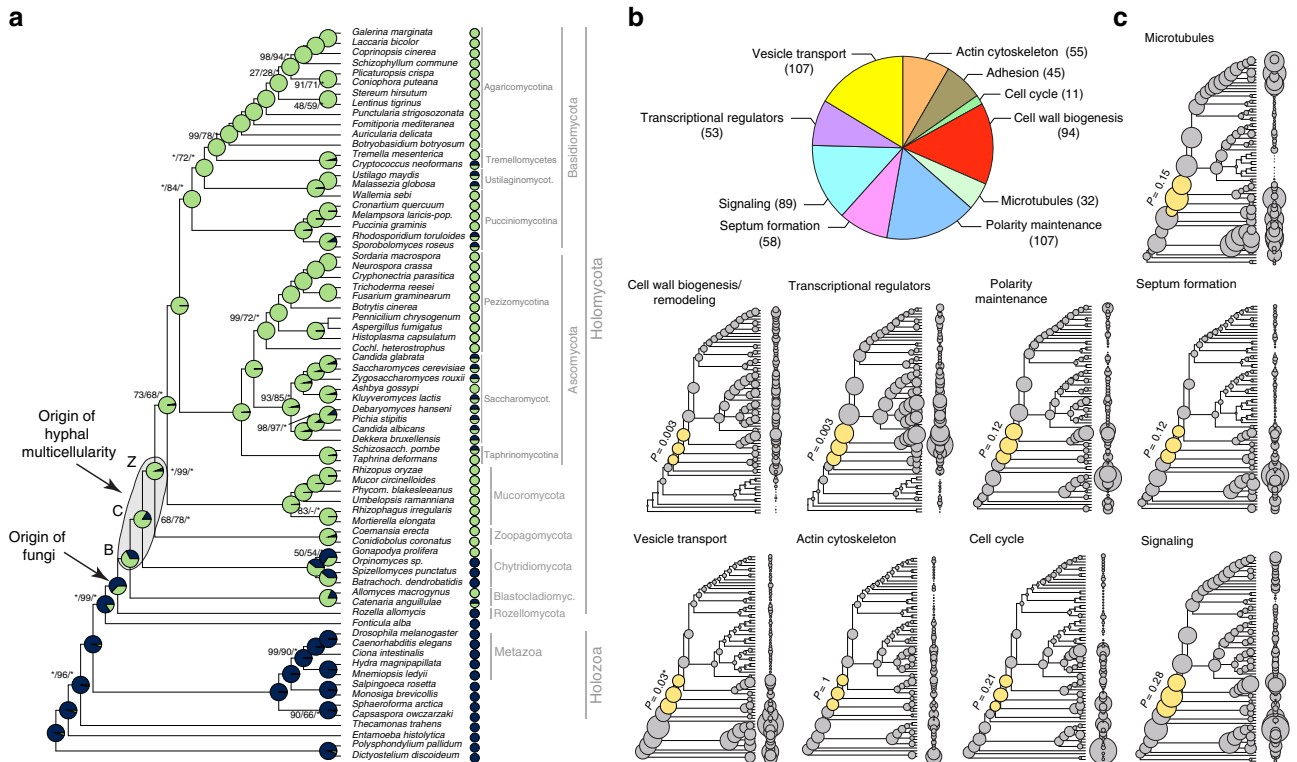

**Fig. 1** The evolution of hyphal multicellularity and underlying genes in fungi. **a** Phylogenetic relationships among 72 species analyzed in this study. Pie charts at nodes show the proportional likelihoods of hyphal (green) and non-hyphal (dark blue) ancestral states reconstructed using Bayesian MCMC. Character state coding of extant species used in ancestral state reconstructions is shown next to species names. BCZ nodes: origins of hyphal growth could be assigned with confidence are highlighted (note the uncertainty imposed by filamentous Blastocladiomycota). Support values next to branches are given for nodes that received less than maximal support in at least one analysis. Support values are given as ML boostrap (RAxML)/ML bootstrap (IQ-Tree)/ Bayesian posterior probabilities (Phylobayes). Asterisk (*) denotes maximal support in a given analysis. **b** the distribution of literature-collected hypha morphogenesis genes among 10 main functional categories. **c** Ancestral reconstructions of gene copy number in 9 main hypha morphogenesis-related categories of genes (see Fig. 2c for adhesion). Bubble size is proportional to reconstructed ancestral gene copy number. BCZ nodes are shown in yellow. P-values of enrichment of duplications are shown next to each tree (Fisher's exact test, FDR correction). *For vesicle transport the P-value indicates significant depletion of duplications in BCZ nodes

morphologies in the Blastocladio- and Chytridiomycota and is consistent with convergent origins of hypha-like morphologies[7,15,16].

To analyze the evolutionary history of putative multicellularity-related genes, we first reconstructed gene family origins and gene duplication/loss histories across all the gene families in the examined genomes (Supplementary Fig. 2). In the following sections, we mine this gene duplication/loss catalog for gene families with previously suggested or novel role in multicellularity and hyphal growth.

**No expansion of kinase, receptor and adhesive repertoires in fungi.** The increased sophistication of cell–cell communication and adhesion pathways in multicellular lineages often correlates with expanded repertoires of genes encoding kinases, receptors and adhesive proteins[53,54]. We therefore first tested if these gene families had undergone diversification in BCZ nodes. Ser/Thr kinase (954 clusters), hybrid histidine kinase (96 clusters), receptor (183 clusters) and adhesion (23 clusters) genes (Supplementary Data 2) did not show expansions reminiscent of patterns in other multicellular lineages (Fig. 2). Ser/Thr kinase repertoires were similar in unicellular and simple multicellular fungi, with higher kinase diversity found in complex multicellular Basidiomycota (as reported by Krizsán et al. 2019)[55] and in *Rhizophagus irregularis* (Fig. 2a). We inferred net contractions in

BCZ nodes, from 572 to 529 reconstructed ancestral kinases (81 duplications, 124 losses, Fig. 2a). Nevertheless, kinase families that duplicated here include all 3 MAPK pathways in fungi, the mating pheromone, cell wall integrity, and osmoregulatory pathways, all of which indirectly regulate hyphal growth[40,56].

Overall, fungi had fewer Ser/Thr kinases (mean 257) than metazoans (mean 643), non-fungal opisthokonts (mean 392), including *Fonticula alba*, the closest relative of fungi (Fig. 2a, Supplementary Note 1). While signal transduction requirements of metazoan MC have been mostly discussed in the context of receptor tyrosine kinases, we found no evidence for domain architectures typical of receptor kinases in fungi. The only group resembling receptor kinases are hybrid histidine kinases (HK), which include a sensor domain, a histidine kinase domain, and a C-terminal receiver domain that acts as a response regulator. We inferred an expansion (24 duplications, 10 losses) of HKs in the MRCA of the Chytridiomycota and other fungi, including class III and X HKs, which are linked to morphogenesis[57,58] (Supplementary Fig. 3). Another wave of HK expansion was inferred in the MRCA of Mucoromycota and Dikarya with 11 duplications and 4 losses.

Canonical G-protein coupled receptors (GPCRs) showed an even more extreme difference between fungi and metazoans (Fig. 2b). We analyzed 183 GPCR families; a large expansion was observed in animals, resulting 135-583 genes in extant species, whereas, only 19 were found in fungi and only one of them

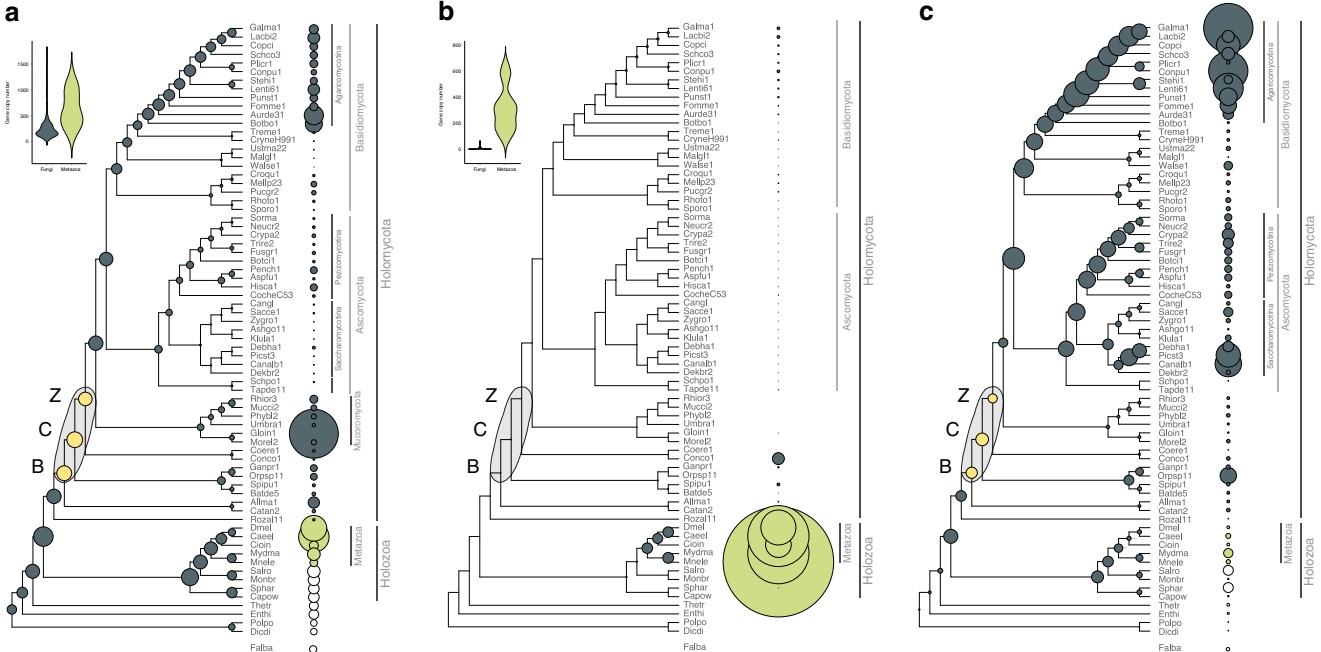

**Fig. 2** The evolution of kinases, receptors and adhesive proteins in multicellular fungi. The evolution of Ser/Thr kinases (**a**), canonical GPCRs (**b**) and adhesion-related genes (**c**). BCZ nodes (yellow) represent the putative origin of hyphal MC. Bubble size across the tree is proportional to reconstructed ancestral gene copy number (gray bubbles) and extant gene copy number (at the right side of the tree: gray, green and white bubbles represent fungi, metazoa, and protists respectively). Violin plots for kinases (**a**) and receptors (**b**) show copy number distribution of gene families in multicellular fungi (gray) and metazoans (green)

(mating pheromone receptors) was conserved across the kingdom.

Adhesive cell surface proteins are key to the emergence of MC in colonial and aggregative lineages[3,5,6], which is reflected in their higher copy numbers in multicellular organisms[59]. We identified 23 families of putative adhesion-related proteins in fungi, including adhesins, flocculins, hydrophobins, various lectins, and glycosylphosphatidylinositol-anchored cell wall proteins. These families have undergone a small contraction (from 17 to 14 copies) in BCZ nodes, with expansions observed later, in the Agaricomycotina and in the Saccharomycotina (Fig. 2c). The lack of an expansion in BCZ nodes probably reflects a marginal role of these proteins in the evolution of early hyphae, but potentially also the scarcity of adhesive proteins annotated in early-diverging fungi, or the effects of sequence divergence. The expansion in the Agaricomycotina was driven by class 1 hydrophobins and homologs of the *Cryptococcus neoformans* Cfl1 (with roles in signaling and morphogenesis regulation[60]) and correlates with the evolution of complex multicellular fruiting bodies[7]. The higher copy numbers in yeast species relate to yeast-specific adhesin and lectin-like cell wall proteins that have been experimentally characterized in human pathogens (e.g., *Candida* spp.)[61,62].

Taken together, the evolution of kinase, receptor, and adhesive protein repertoires highlight an important difference between fungi and other multicellular lineages. We observed no significant expansion of these families in filamentous fungi, whereas kinase and adhesion-related genes expanded in complex multicellular Agaricomycotina. This might be explained by the two-step nature of the evolution of complex MC in fungi[7,63] that proceeds through an intermediate complexity level, hyphal MC, as opposed to metazoans, where complex MC evolved in a more direct way[13]. The observation that these 'classic' culprits of multicellular evolution can't explain the evolution of hyphae prompted us to

examine other gene families, whose evolution might show a better correlate with that of hyphae.

**The evolution of hypha morphogenesis genes.** We built a dataset of hypha morphogenesis genes to determine whether changes in these gene families correlate with the evolution of hyphae. We identified 651 hyphal multicellularity-related genes belonging to 362 families (from 519 publications, covering our current knowledge on hyphal growth)—mostly derived from well-studied model systems such as *A. fumigatus*, *A. nidulans*, *N. crassa*, *S. cerevisiae* and *C. albicans* (Supplementary Data 3). We categorized genes into nine functional groups according to the broader function they serve in hyphal growth: actin cytoskeleton regulation, polarity maintenance, cell wall biogenesis/remodeling, septation (including septal plugging), signaling, transcriptional regulation, vesicle transport, microtubule-based transport and cell cycle regulation. The categories "polarity maintenance" and "vesicle transport" contained the largest number of genes (107 in each), whereas "cell cycle regulation" contained the fewest (11) (Fig. 1b). To account for uncertainty in the exact origin of hyphae, we hereafter focus on BCZ nodes in our analyses of hypha morphogenesis genes. We examined multiple types of evolutionary innovations, to identify the most important mechanisms underlying the emergence of hyphae.

Reconstructions of gene duplication/loss histories for the nine functional categories of hypha morphogenesis gene families are shown on Fig. 1c. A general pattern that emerges from these is that most of the gene families are conserved across fungi (Supplementary Fig. 4) and their origin predate that of hyphae (181 families, 50%, Fig. 3, Supplementary Fig. 5), indicating that fungi have co-opted several conserved eukaryotic functionalities for hyphal growth. Gene families related to septation, polarity maintenance, cell cycle control, vesicle

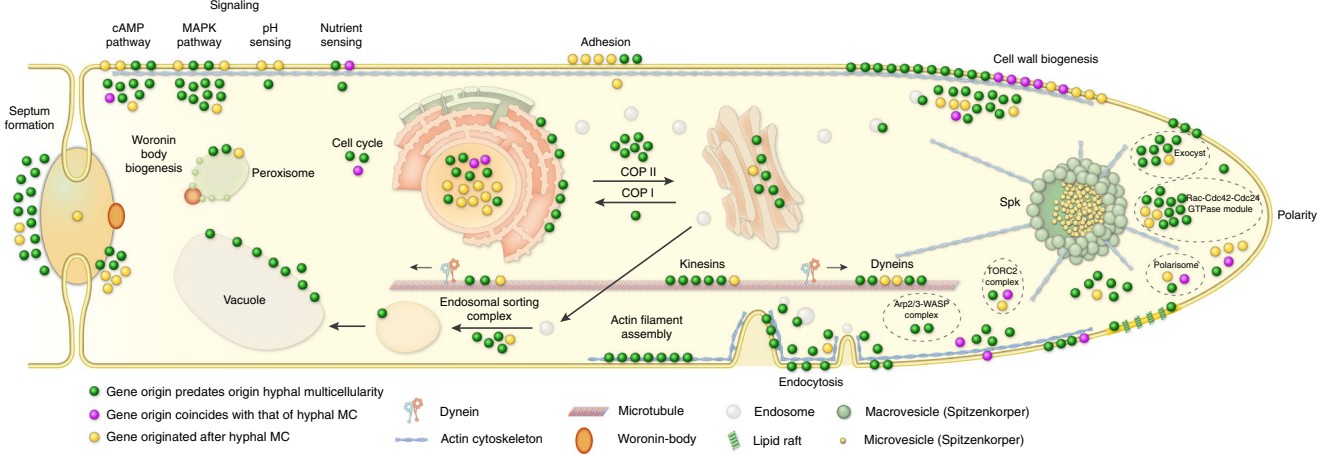

**Fig. 3** Phylogenetic age distribution of hypha morphogenesis genes. Schematic outline of terminal hyphal cell is shown with genes marked by dots and colored by phylogenetic age. Genes whose origin (based on their containing gene family) predates that of hyphal multicellularity (green) dominate the hypha morphogenetic machinery, followed by genes that originated after hyphal MC (yellow) and genes whose origin approximately coincides with that of hyphae (purple). Data based on only *A. fumigatus* orthologs. See Supplementary Fig. 5 for gene names

transport, and microtubule-based transport are generally more diverse in animals, non-fungal eukaryotes (including *Fonticula alba*, see Supplementary Note 1) and their ancestral nodes than in fungi, suggesting that despite the key role of these families in hyphal MC, they evolved primarily by gene loss in fungi (Fig. 1c). A significant proportion of hypha morphogenesis families (164 families, 45.3%) emerged after the origin of hyphal MC, indicating lineage- and species-specific genetic innovations. Only 17 families (4.7%) originated in BCZ nodes and were conserved thereafter (Table 1), providing potential candidates that shaped the evolution of hyphal MC. These include two families of transcriptional regulators (StuA and MedA in *A. fumigatus*), six related to cell wall biogenesis, three to actin cytoskeleton regulation, three to polarity maintenance, two families involved in signaling and one involved in cell cycle regulation. These fungal-specific families are good candidates for being key contributors to the evolution of multicellular hyphae; their functions are summarized in Table 1.

Given the low number of gene families specific to multicellular fungi, we were interested in whether evolutionary innovation by duplications shows a peak in BCZ nodes. Ninety-three (25.7%) of the 362 hypha morphogenesis-related gene families showed duplications in BCZ nodes (Supplementary Data 4). Enrichment analyses, however, revealed no individual gene family with significantly increased number of duplications in BCZ nodes relative to the rest of the tree (Benjamini–Hochberg corrected $P <$ 0.05, Fisher's exact test, Supplementary Data 5). The same analysis on the 9 functional groups showed that duplications are significantly enriched in cell wall biogenesis and transcriptional regulator genes, suggesting that their diversification could have played roles in the evolution of hyphae. The rest of the functional groups showed no such enrichment of duplications, suggesting that the evolution of hyphal growth did not generally coincide with a major burst of gene duplication in BCZ nodes, as some considerations of the evolution of multicellularity predicted[27].

Changes in structural properties of genes show a correlation with the evolution of hyphal MC. Significant differences ($P < 0.05$, two-tailed Welch's *t*-test with pooled variance estimation) were observed in gene, coding sequence (CDS) and intron lengths between unicellular and multicellular fungi in 7 out of 9 functional groups (exceptions are cell wall biogenesis and transcriptional regulation genes) (Supplementary Fig. 6, Supplementary Data 6). Coding sequences of septation and polarity maintenance genes were significantly longer in multicellular than

in unicellular fungi ($P = 0.0012$-$0.00017$, Supplementary Fig. 6). An opposite pattern was observed in introns, which were on average longer in unicellular fungi in actin cytoskeleton, polarity maintenance, septation and vesicle transport-related genes. On the other hand, no significant changes in gene structure were detected in cell wall biogenesis and transcriptional regulation-related genes, the two categories that displayed increased duplicability in early filamentous fungi. This depicts potential complementary mechanisms of evolutionary change in different gene families and functional groups of genes.

We also analyzed changes in domain composition across plesiomorphically unicellular and multicellular fungi. This analysis was inspired by Class V and VII chitin synthases, which evolved higher efficiency in filamentous fungi by gaining a myosin motor domain during evolution[64,65]. Hypha morphogenesis gene families, in general, show more change in domain composition between unicellular and filamentous fungi than do randomly drawn gene families with similar properties (Supplementary Fig. 7). This indicates that changes to domain architectures correlate with the emergence of hyphae. We identified 4 gene families (including chitin synthases), in which proteins of multicellular fungi have consistently more domains ($P < 0.05$, GLM) than do proteins of unicellular fungi in the same family (Supplementary Data 7). Taken together, these analyses revealed several modifications to gene length and domain composition in multicellular fungi, which, although individually are small changes, could have contributed to the evolution of hyphae.

**Phagocytotic genes were exapted for hypha morphogenesis.** Our set of hypha morphogenesis genes included several entries associated with phagocytosis in non-fungal eukaryotes. This is surprising given that phagocytosis is not known in fungi and their rigid cell wall forms a physical barrier to it. We, therefore, examined the fate of phagocytosis genes in filamentous fungi based on the phagocytotic machinery of *D. discoideum*[66,67] and other eukaryotes[68] (altogether 106 genes). Filamentous fungi have retained several phagocytotic gene families but lost others (Fig. 4). For example, members of the Arp2/3 complex (involved in actin cytoskeleton rearrangements[69]) are conserved in filamentous fungi and recycle excess membrane in the subapical zone during hyphal growth[70]. Engulfment and cell motility (ELMO1/2) genes are found in all filamentous fungi, but are convergently lost in

**Table 1 List of the 17 gene families whose emergence coincides with the evolution of hyphae**

| Emergence of gene family | *A. fumigatus* ortholog | *S. cerevisiae* ortholog | Functional category and putative function |
|---|---|---|---|
| mrca of Dikarya, Mucoromycota, Zoopagomycota, Chytridiomycota, Blastocladiomycota | Afu7g03870 | PAN1 | Actin cytoskeleton: endocytic adaptor that triggers hyphae-specific recruitment of the Arp2/3 complex to sites of endocytosis, for the recycling of excess membrane in the subapical region during hyphal growth[41] |
| | crh3 | UTR2 | Cell wall biogenesis: chitin transglycosylase, localized to sites of polarized growth, functions in the transfer of chitin to beta(1–6) and beta(1–3) glucans |
| | gel7 | GAS1 | Cell wall biogenesis: beta(1–3) glucanosyltransferase, involved in cell wall remodeling during fungal germination or branching |
| | Afu6g04940 | BNR1 | Polarity establishment: mediates actin cable assembly in filamentous fungi and has a role in diverse morphogenetic processes[72] |
| | Afu4g04120 | BEM1 | Polarity establishment: actin cytoskeleton reorganizing factor[73,74] |
| | stuA | PHD1 | Transcriptional regulation: mediates yeast–filament transition in *S. cerevisiae*, developmental modifier in *A. fumigatus*, that spatially and temporally regulates the central transcription factor cascade |
| | medA | NA | Transcriptional regulation |
| mrca of Dikarya, Mucoromycota, Zoopagomycota, Chytridiomycota | Afu6g07910 | SLM1 | Actin cytoskeleton: effector of PtdIns(4,5)P2, essential for cell growth and actin cytoskeleton polarization |
| | Afu8g04520 | SLA1 | Actin cytoskeleton: actin cytoskeleton-regulatory complex protein, localized to the actin patches that form the sites of endocytosis |
| | Afu4g06130 | WHI2 | Cell cycle regulation: required for cell cycle regulation and stimulates filamentous growth |
| | Afu4g00620 | DFG5 | Cell wall biogenesis: mannosidase, involved in bud formation and filamentous growth |
| | Afu8g02320 | NA | Cell wall biogenesis: ortholog of *N. crassa* cps1 polysaccharide synthase, functions in cell wall biosynthesis |
| | chsD | NA | Cell wall biogenesis: class VI chitin synthase, role in chitin biosynthesis |
| | rgsB | RAX1 | Polarity establishment: bipolar budding in *S. cerevisiae*[75] |
| | Afu2g08800 | SSY1 | Signaling: component of the SPS plasma membrane amino acid sensor system |
| | ricA | NA | Signaling: GDP/GTP exchange factor for G proteins, role in regulating fungal development |
| mrca of Dikarya, Mucoromycota, Zoopagomycota | kre6 | KRE6 | Cell wall biogenesis: role in beta(1–6) glucan biosynthesis |

budding and fission yeasts, in *C. neoformans*, *M. globosa* and *W. sebi*, all of which have reduced capacity for hyphal growth. The DOCK (dedicator of cytokinesis, *S. cerevisiae* DCK1[71]) protein family, which interacts with ELMO proteins, is retained in fungi. Of the broader Wiskott–Aldrich syndrome family of proteins, which reorganize the actin cytoskeleton during phagocytosis, the WASP family is conserved across fungi, the WAVE family is only represented in early diverging fungi and non-fungal eukaryotes, whereas the WASH family has been completely lost in fungi (Fig. 4), consistent with recent reports[72,73]. These patterns reveal the conservation of several phagocytic genes in fungi, despite the loss of phagocytosis itself. This highlights exaptation as another mechanism for the recruitment of genes for hyphal growth.

**Genome-wide screen finds novel gene families linked to hyphae.** We next asked if there were further gene families that have a potential connection to the evolution of hyphal MC. We reasoned that gene families underlying hyphal MC should originate or diversify in BCZ nodes and be conserved in descendent filamentous fungi. A systematic search for gene families fitting these criteria yielded 414 families (ANOVA, $P < 0.05$,

Supplementary Data 8), 114 of which originated in BCZ nodes, while the others showed duplication rates that exceeded the expectation derived from genome-wide figures of gene duplication (Fig. 5). The conservation of these putative hyphal multicellularity-related gene families across fungi is shown on Fig. 5. These families included several known morphogenetic families (e.g. Bgt3, RgsB and Gel2 of *A. fumigatus*, Bem1 and Rax1 of *S. cerevisiae*), genes involved in actin cytoskeleton and cell wall assembly, mating, pheromone response (GpaA of *A. fumigatus*), sporulation and transport, among others (Supplementary Data 8). Several of the identified families contain genes with reported growth defects in *A. fumigatus* or *S. cerevisiae*, indicating that our searches recovered genes relevant for hyphal MC. For example, Rax proteins are major regulators of cellular morphogenesis and are involved in bud site selection in budding yeasts[74], polarized growth in *S. pombe*[75] and polarity maintenance in filamentous fungi[76]. We further detected a fungal-specific cluster of tropomyosins (TPM1 in *S. cerevisiae*), which originated in the MRCA of Blastocladiomycota and other fungi and is involved in polarized growth and the stabilization of actin microfilaments. The family containing *S. pombe* Dip1 homologs (Afu6g12370 in *A. fumigatus*) emerged in the node uniting Chytridiomycota with higher fungi and contains a single gene per

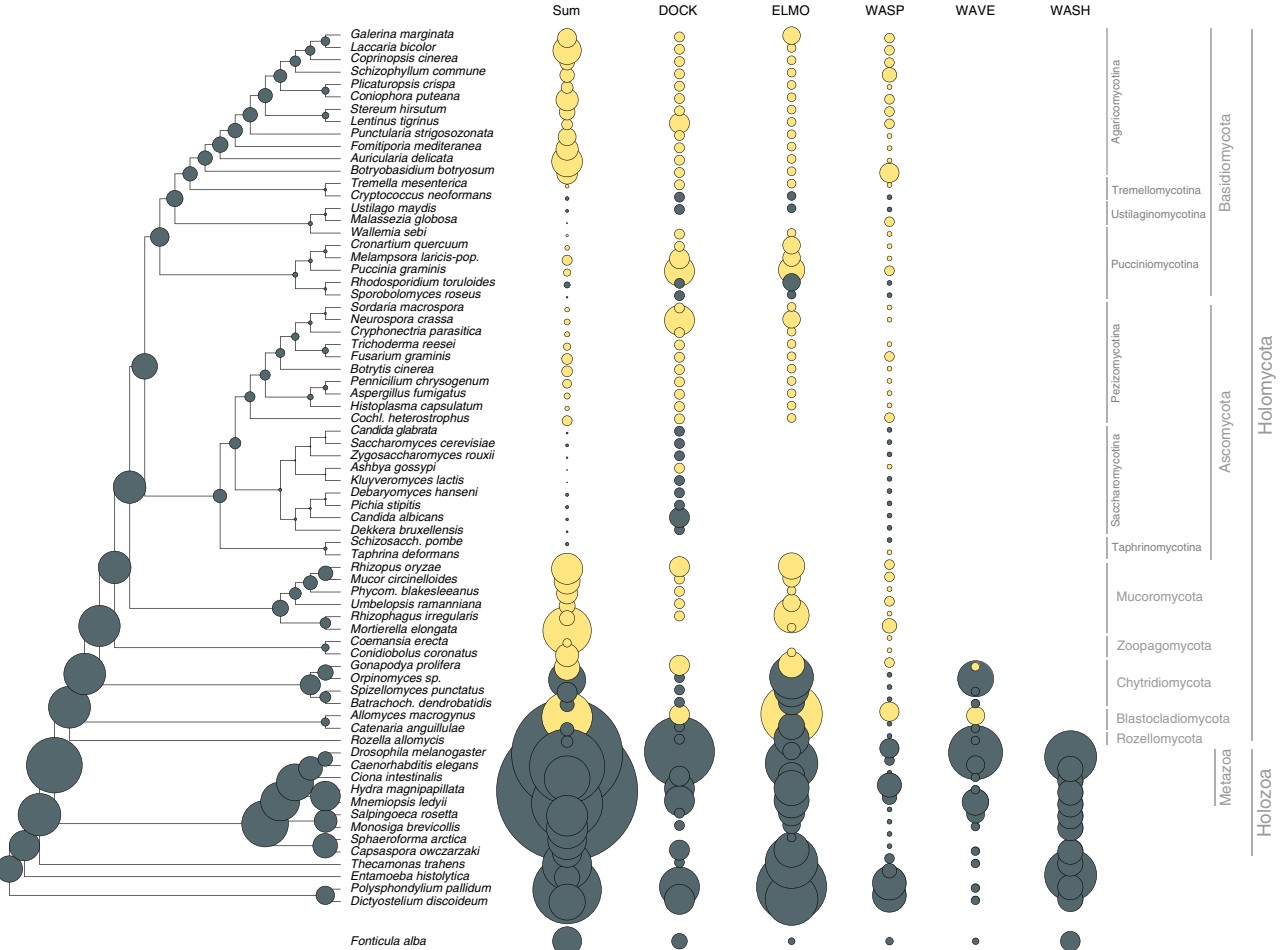

**Fig. 4** Evolutionary history of phagocytosis-related gene families. Several phagocytotic gene families retained in filamentous fungi (DOCK, ELMO, WASP). WAVE family retained only in early fungi (Blastocladiomycota and Chytridiomycota), WASH family is represented only in non-fungal eukaryotes. Bubble size is proportional to ancestral and extant gene copy number. Copy numbers of filamentous fungi are labeled with yellow

species afterward, except an expansion in WGD Mucoromycota[77] and losses in the Saccharomycotina. In *S. pombe*, Dip1 activates the Arp2/3 complex without preexisting actin cables[78] and thus regulates the actin cytoskeleton through a mechanism that seems to be specific to multicellular fungi. Finally, we detected the family containing *S. cerevisiae* Dpp1 homologs, which regulate morphogenetic transitions in dimorphic fungi through the synthesis of the fungal signal molecule farnesol[79], which prompts us to speculate that it might have contributed to the elaboration of farnesol-based communication in fungi. Collectively, the origin of these families in BCZ nodes makes them candidate key contributors to the evolution of hyphal MC.

Because the emergence of hyphal multicellularity overlaps with that of other fungal traits, it is challenging to unequivocally separate signals conferred by the emergence of these traits from those by hyphae. It is conceivable that a portion of the 414 gene families were detected because of signals conferred by phylogenetically co-distributed traits, not necessarily multicellularity itself (see Beaulieu 2016[80] for a conceptually analogous problem in taxonomic diversification). One such trait could be osmotrophy, feeding by the absorption of soluble 'public goods' generated by the activity of secreted extracellular enzymes[81]. We detected 20 gene families that showed strong correlation with hyphal MC and were annotated as various transporters; such families could conceivably be related to osmotrophy. Further, there were 84 families that are functionally uncharacterized and thus it is impossible to speculate about their role in hyphal MC. These

families suggest that there are many fungal genes that evolved in concert with hyphal MC and await functional characterization to understand their roles.

**Yeasts retain genes required for hypha morphogenesis.** Yeasts are secondarily simplified organisms with reduced ability to form hyphae, that spend most of their life cycle as unicells[16,22,46,82]. Our ancestral character state reconstructions imply that yeasts derived from filamentous ancestors (Fig. 1a), and thus they represent a classic example of reduction in complexity. They were hypothesized to have lost MC[83], even though rudimentary forms of hyphal growth (termed pseudohyphae) exist in most species. We scrutinized the fate of MC-related genes in five predominantly yeast-like lineages[82] (Fig. 6). Because yeast genomes have undergone extreme streamlining during evolution, we evaluated gene loss among hypha morphogenesis genes in comparison to genome-wide figures of gene loss.

Yeast species generally have fewer hypha morphogenesis genes and reconstructions indicate more losses than duplications along branches of yeast ancestors (Fig. 6). However, when we corrected for genome-wide reductions in gene number, we found that comparatively fewer hypha morphogenesis genes (35–46%) were lost than genes with other functions (46–59%, Fig. 6a, Supplementary Data 9). Gene loss in yeast clades is significantly depleted ($P < 0.05$, Fisher's exact test) in most groups of hypha morphogenesis genes compared to genome-wide expectations.

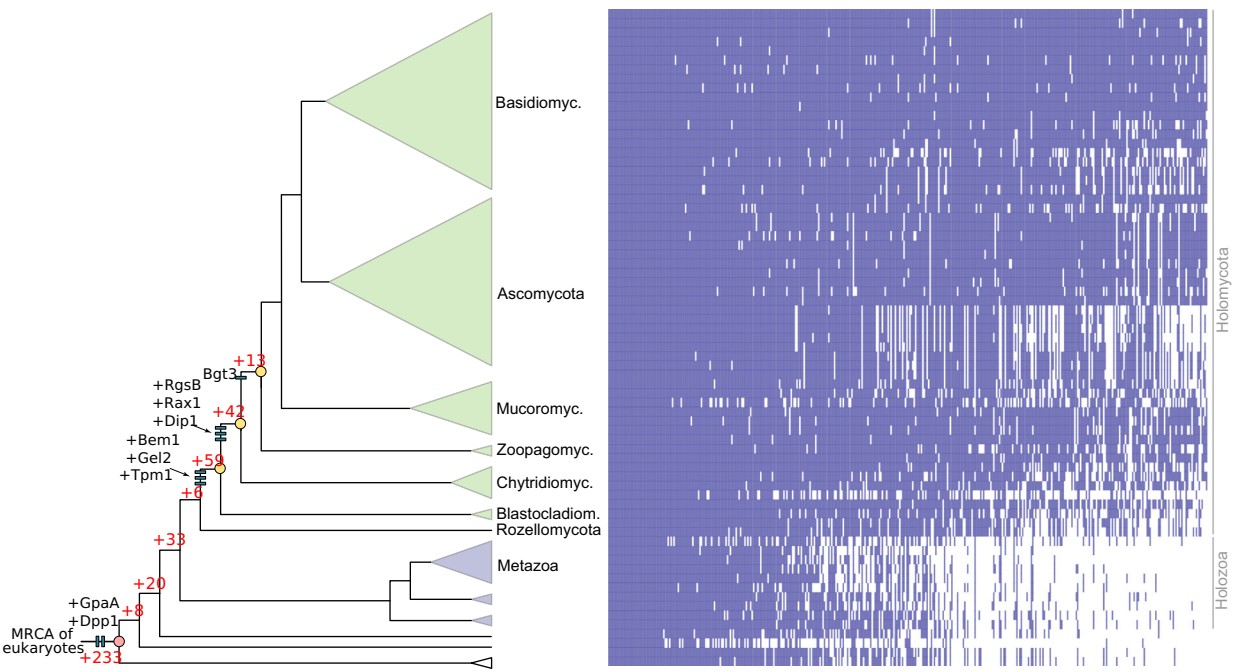

**Fig. 5** Origin of 414 gene families potentially related to the evolution of hyphal MC, identified by ANOVA ($P < 0.05$). 114 families originated in BCZ nodes (shown in yellow), including known morphogenesis-related proteins (e.g. Bgt3, RgsB, Gel2 of *A. fumigatus*, Rax1, Bem1, Tpm1, and Dpp1 from *S. cerevisiae*, Dip1 from *S. pombe*) labeled as blue bars. Red numbers represent the number of gene families originated at the respective branches. Heatmap next to the tree shows the conservation of the identified 414 gene families across fungi. Blue and white colors indicate that the gene family is present or absent in the genome, respectively

We recovered only six cases where significantly more MC-related genes were lost than expected ($P < 0.05$, Fisher's exact test, Supplementary Data 9). These included proteins related to adhesion and microtubule-based transport, suggesting that these functions are generally dispensable for yeast clades. Contractions (e.g. gamma-tubulin complex, kinesins, dynein heavy chain (nudA)) or complete losses (e.g., dynactin, gcpC, and the dynactin linking protein ro-2[84]) of gene families in microtubule-based transport are particularly interesting from the perspective of long-range transport of vesicles and nuclei along hyphae. The budding and fission yeast lineages show the most losses in these genes, consistent with the strongest reduction of hyphal growth abilities in these clades. Losses of NADPH oxidases were observed in all yeast clades, with complete loss of the family in the Saccharomycotina, Ustilaginomycotina, in *S. pombe* and *C. neoformans* (Supplementary Fig. 8), as reported previously[85].

Collectively, these analyses suggest that hypha morphogenesis genes are, in general, less dispensable for yeasts than genes with other functions. This agrees with most yeast-like fungi being able to switch to hyphal or pseudohyphal growth under certain conditions. The inferred gene losses, nevertheless, do indicate reductions in hyphal growth, but this reduction is smaller than that of other functions in the genome. This, in turn means, that the ability for multicellular growth is among the functions preferentially retained by yeast-like fungi.

## Discussion
In this study, we analyzed the genetic underpinnings of the evolution of fungal hyphae, the most enigmatic of fungal structures, with a unique multicellular organization but a poorly understood evolutionary origin. Our analyses suggest that hyphae evolved in early fungi (the split of Blastoclado-, Chytridio- and Zoopagomycota, termed BCZ nodes), consistent with the

previous studies[15]. To understand how the underlying genetics evolved, we reconstructed the evolution of 362 hypha morphogenesis gene families and predicted a link to hyphae for another 414 families.

The evolutionary picture developed from these analyses has several conspicuous features. Most families were conserved across all sampled eukaryotes with few or no duplications at the origin of multicellular fungi. A second category of gene families show a deep eukaryotic origin and duplications coincident with the evolution of hyphae (e.g., cell wall biogenesis and transcriptional regulation-related genes). However, none of these families had a significantly elevated duplication rate in BCZ nodes, indicating limited innovation via gene duplication. A third category consisted of gene families whose origin map to BCZ nodes. These could have evolved de novo or they could have diverged so much that similarity to homologous non-fungal sequences cannot be detected. We find support for both of these possibilities. For example, the MedA or APSES families contain fungal-specific protein domains; these have conceivably evolved in early fungi and represent fungal-specific innovations underlying hyphal growth. On the other hand, the detected formin and RGS families contained only fungal genes, but their characteristic Interpro domains occur outside of fungi too, possibly reflecting common ancestry, with evidence for homology blurred by sequence divergence.

Beyond gene family events, our analyses revealed that changes in the length of genes/introns and domain architectures of MC-related genes in multicellular vs unicellular fungi also correlate with the emergence of hyphae. There is also evidence for changes to amino acid sequence, for example in fungal kinesins that are 2× more processive than other eukaryotic kinesins[86], conceivably as a result of selection for efficient long-range transport along the hyphal axis.

Co-option and exaptation may have been the most important source of genes for hyphal MC, followed by gene duplications,

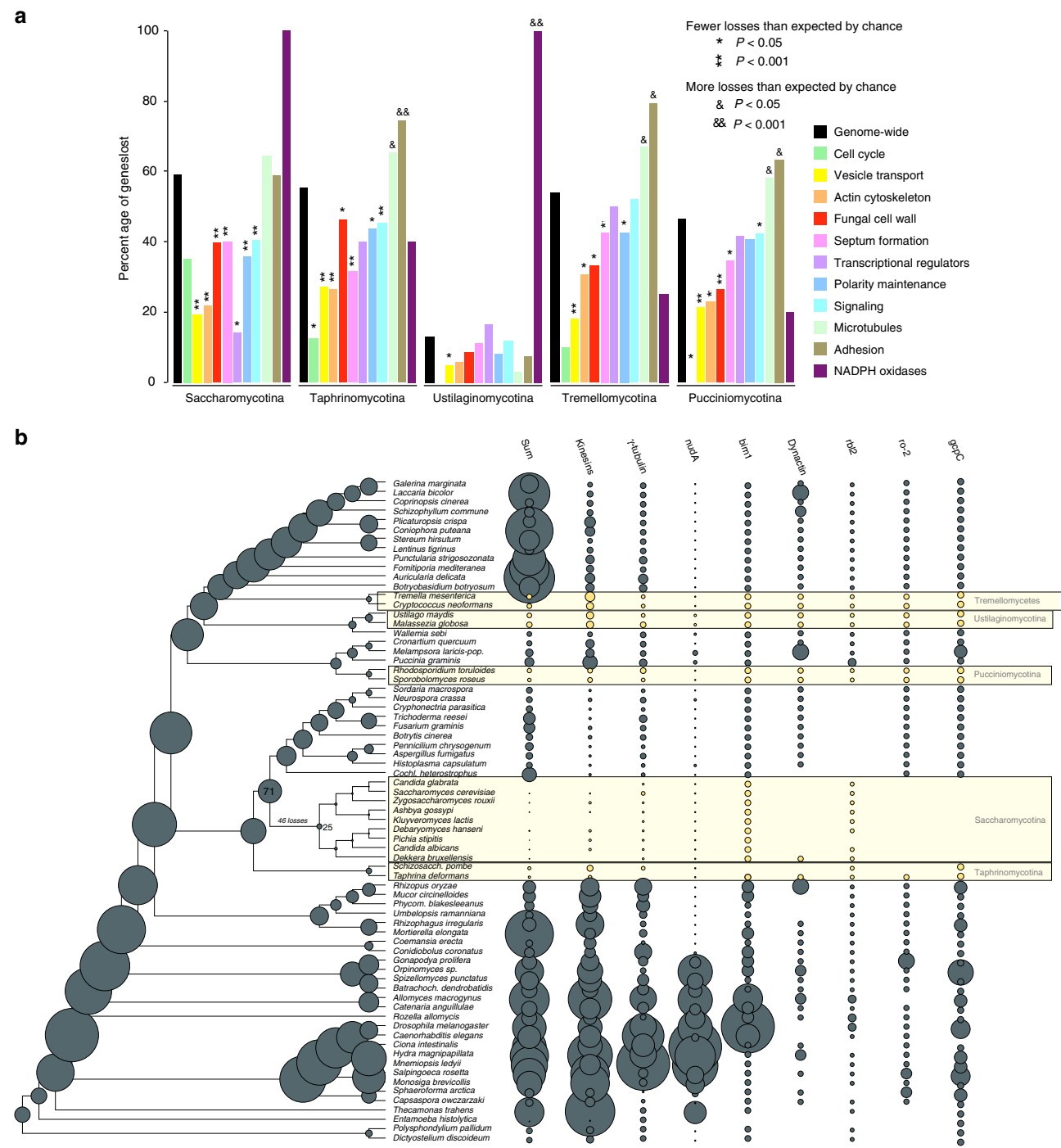

**Fig. 6** Secondarily simplified yeast-like fungi retain genes for hyphal MC. **a** the percentages of lost genes in main morphogenesis-related categories. Percentages were calculated relative to ancestral copy numbers inferred in the node preceding the origin of 5 yeast-like clades (Saccharomycotina, Taphrinomycotina, Pucciniomycotina, Ustilaginomycotina, and Tremellomycotina). Significance of the enrichment of gene losses in each category relative to genome-wide figures of gene loss were determined by Fisher's exact test and is shown above bars. (**b**) ancestral gene copy number reconstruction of microtubule-based transport genes along the fungal phylogeny. Secondarily simplified (yeast-like) clades are highlighted in yellow. Bubble size proportional to reconstructed ancestral and extant gene copy number across 19 gene families. Copy number distribution of each gene family is shown right to the tree

structural changes to genes and de novo gene family birth. These findings mirror patterns of the evolution of multicellular animals and plants (e.g. refs. [6],[87]–[89]) that gave rise to the hypothesis that, in terms of genetic novelty, transitions to multicellularity represent a minor rather than a major evolutionary step[90], an idea that finds support in the observations made here on fungi.

Our observations highlight how a wealth of small genetic changes may synergistically lead to a key evolutionary innovation, such as fungal hyphae. This 'tinkering'[48] process is consistent with the stepwise, gradual evolution of hyphae and could explain why transitional hypha-like forms exist in early-diverging fungi (Blastocladiomycota, Chytridiomycota)[15]. The dynamic cytoskeletal and endomembrane systems have been suggested to underlie

morphological patterning in eukaryotes[91]. Based on the co-option of such genes for hyphal growth, we speculate that the cytoskeleton was key to the evolution of hyphae, along with other gene families and innovations. Apart from the foregoing mechanisms, gene losses in other families, exaptations and concurrent fungal innovations (e.g. osmotrophy or the cell wall), might have eventually led to hyphae being the optimal solution for multicellularity in fungi. Hyphal growth, in turn, paved the way for diverse cell differentiation programs in fungi, which might have been a prerequisite to the repeated emergence of complex multicellular structures later in evolution[7].

Compared to other multicellular lineages, the evolution of multicellular fungi shows several unique patterns. While the expansion of adhesion and signal transduction mechanisms is shared by colonial and aggregative multicellular lineages examined so far[9,17,92,93], we did not find evidence for this in multicellular fungi. This could be explained by the apical, fractal-like growth mode of fungal hyphae, which shares similarity only with the Oomycota. While adhesion might not be key in vegetative hyphae, there is plenty of evidence for active communication between neighboring hyphae[94,95]. It is possible that the main modes of communication in fungal hyphae are not linked to canonical cell surface receptors (see e.g., ref. [96]), but instead are, either mediated by volatiles (such as farnesol[79,95,97]) or are not known yet. These observations suggest that multicellularity in fungi differs considerably from that in other lineages and raises the possibility that in addition to the aggregative and clonal modes of evolving MC, fungal hyphae should be considered a third, qualitatively different route to evolving multicellularity. Subjective categorizations aside, hyphae represent a highly successful adaptation to terrestrial life and comparative genomics opens the door for dissecting the genetic underpinnings of its emergence and for discussions on whether such, major phenotypic innovations represent—in terms of genetic novelty—a major or a minor transition.

## Methods

**Organismal phylogeny.** We assembled a dataset containing whole proteomes of 71 species plus *Fonticula alba* (which was published after assembling the main dataset and was thus added later) and performed all-vs-all blast using mpiBLAST 1.6.0[98]. Taxon sampling was designed so as to cover all major clades of fungi (following the classification of Spatafora et al.[52]), irrespective of their cellularity status, to avoid biasing the dataset towards hyphal species. Accordingly, we selected species from all three subphyla of both the Asco- (Taphrino-, Saccharo- and Pezizomycotina) and of the Basidiomycota (Agarico-, Puccinio- and Ustilaginomycotina), as well as a representative set of Mucoromycota, Zoopagomycota, Chytridiomycota, Blastocladiomycota. and Cryptomycota. We omitted Microsporidia from the dataset due to the high rate of evolution of this group. Multicellular animals, their unicellular relatives, and other non-fungal eukaryotes were included as outgroups and for comparison of some multicellularity-related gene families across kingdoms. Proteins were clustered into gene families based on sequence similarity by following the MCL clustering protocol[99] used by Ohm et al.[100] with an inflation parameter of 2.0 (Supplementary Data 10). Clusters with at least 50% taxon occupancy were chosen and were aligned by PRANK 140603[101] while trimAl 1.4.rev15[102] was used to remove poorly aligned regions from the multiple sequence alignments using the parameter –gt 0.2. Approximately-maximum-likelihood gene trees were inferred by FastTree[103] using the LG+CAT model (-lg -cat20), and the option -gamma to compute a Gamma20-based likelihoods. We excluded gene families that contained deep paralogs by analyzing gene duplication patterns in inferred gene trees (see ref. [104] for details). Alignments of single-copy gene clusters were concatenated into a supermatrix (Supplementary Data 11) and we inferred a species tree for the 71 species in RAxML 8.2.4[105] under the PROTGAMMAWAG model (Supplementary Data 12). The model was partitioned by gene and bootstrapping was performed on the dataset in 100 replicates. Second, we used the more sophisticated models implemented in IQ-TREE v1.6.10[106,107], again with the model partitioned by gene and 100 conventional bootstrap replicates (Supplementary Data 13). The optimal model for each partition was determined by ModelFinder[108], as implemented in IQ-TREE. Model selection was restricted to homogeneous LG and two protein mixture models (LG4X and LG4M) allowing empirical frequencies with discrete gamma rate heterogeneity or the FreeRate model. We also used Bayesian MCMC under the CAT model as implemented in PhyloBayes[109] to reconstruct tree topology and estimate

clade posterior probabilities (Supplementary Data 14). For this, we ran 3500 cycles (equivalent to ca. 400,000 generations) under the CAT-GTR model in three replicates. Burn-in was determined by examining parameter convergence and runs were considered to have converged to the same posterior the maxdiff parameter of the bpcomp function dropped below 0.01[109].

**Ancestral character state reconstructions.** The 71 species were coded for their ability to form hyphae, either as hyphal or non-hyphal. Species that could not be unambiguously assigned to hyphal or non-hyphal (*Catenaria anguillulae*) and those with the ability to grow either as hyphae or unicells (most yeasts) were coded as uncertain. Bayesian MCMC reconstruction of ancestral character states was performed under the threshold model[110] using Bayesian MCMC with the "ancThresh" function in phytools v0.6-60[111] in R[112]. The number of generations for MCMC was set to 1,000,000, and the method "mcmc" was used with the Brownian motion as the model for the evolution of the liability. Burn-in parameter was set to default. Convergence was checked by inspecting likelihood values through time.

**Analyses of gene family evolution.** To investigate the evolutionary history of gene families containing hypha morphogenesis-related genes, we analyzed clusters that contained such genes based on the above-described clustering of the proteomes of the 71 analyzed species. The resulting protein clusters were aligned by PRANK 140603[101] with default parameters, and ambiguously aligned regions were removed using trimAl 1.4.rev15[102] with the argument –gt 0.2. MAFFT v7.222[113] (option --auto) was used as an alternative alignment tool for clusters that could not be aligned by PRANK due to computational limitations (80 out of 34032 clusters). Maximum Likelihood inference of gene trees and calculation of Shimodaira-Hasegawa-like branch support values were carried out in RAxML 8.2.4[114] under the PROTGAMMAWAG model of protein evolution. The calculated SH-like branch support values were used in gene tree-species tree reconciliation in Notung-2.9[115]. An edge-weight threshold of 0.9 was used, as SH-like support values are usually less conservative than ML bootstrap values (where 70% is usually taken as an indication of strong support). Reconciliation was performed on the maximum likelihood gene trees (gene trees are provided as Supplementary Data 15–16) and the ML species tree for the 71 species as input. We reconstructed the gene duplication/loss dynamics of gene families along the species tree using the scripts for ortholog coding and Dollo parsimony mapping from the COMPARE pipeline[82,116] (available at https://github.com/laszlognagy/COMPARE and in Supplementary Data 17). The numbers of gains and losses for each gene family and for each branch of the species tree were recorded and mapped on the species tree. Ancestral gene copy numbers were calculated for every internal node, summing the mapped duplications and losses across the species tree. Mappings were generated for each of the functional groups and also for kinases, adhesion-related proteins, receptors as well as for all gene families across the 71 genomes.

To test if genes related to hyphal MC experience an episode of increased duplication rate in nodes where hyphal growth putatively originated (BCZ nodes), we performed gene duplication enrichment analysis for each of the 362 families and for functional groups. To test if a cluster or a functional group shows significantly more or less duplications than expected by chance in BCZ nodes, we run two-tailed Fisher's exact tests ($P < 0.05$). We compared the number of duplications mapped to BCZ nodes for a given gene family to the genome-wide number of duplications in BCZ nodes, using total number of duplications across the tree as a reference.

**Analyses of key multicellularity-related genes.** The above strategy was used to reconstruct the evolution of kinase, receptor, and adhesion-related gene families. Protein clusters containing kinase genes, both serine-threonine kinases and histidine kinases, were collected based on InterPro domains. Identification of serine-threonine kinases and histidine kinases followed Park et al.[117] and Herivaux et al.[58], respectively. Classification of histidine kinases followed Defosse et al.[57]

Families of adhesive proteins were identified based on experimentally characterized genes collected from the literature. We identified 45 genes, which mostly grouped into flocculins, lectins, hydrophobins, and other (GPI)-modified cell wall adhesins. We identified receptor genes based on InterPro domains that are annotated with the gene ontology term 'receptor activity' (but not 'receptor binding' or other terms indicative of indirect relationships to receptor function), resulting in 27 IPR terms (IPR000161, IPR000276, IPR000337, IPR000363, IPR000366, IPR000481, IPR000832, IPR000848, IPR001103, IPR001105, IPR001499, IPR001546, IPR001946, IPR002011, IPR002185, IPR002280, IPR002455, IPR002456, IPR003110, IPR003292, IPR003980, IPR003982, IPR005386, IPR006211, IPR017978, IPR017979, IPR017981).

**Analyses of phagocytosis-related genes.** We collected information on phagocytosis-related genes from recent reviews on *Dictyostelium*[66,67], identified the corresponding genes of this species in our dataset and the protein clusters that contained homologs of the identified genes. Mapping of gene duplications and losses along the species tree was done as described above.

**Genome-wide screen for novel hyphae-related gene families**. To identify gene families with increased rates of gene duplication coinciding with the origin of hyphal MC, we set up a pipeline that tests for higher than expected rate of duplication in nodes of the species tree to which the origin of hyphae could be located (BCZ nodes). For each gene family, gene duplication rates in BCZ nodes were compared to duplication rates of the same family in other parts of the species tree (nodes before and after BCZ nodes). Gene duplication rates were computed by dividing the number of reconstructed duplications for a given branch by the length of that branch using a custom Perl script (see Supplementary Data 17). Terminal duplications and duplications mapped to metazoan ancestors were excluded from the analysis. The resulting node × duplication rate matrix was analyzed by a two-factor permutation ANOVA[118] with degrees of freedom DFT = 2, in R, with $P < 0.05$ considered as significant. We further required that the detected clusters be conserved (>=1 copy) in at least 70% of filamentous fungi.

**Analyses of gene losses in yeast-like fungi**. We analyzed gene losses in five yeast-like lineages by comparing the number of losses genome-wide, to the numbers of losses in hypha morphogenesis related genes (actin cytoskeleton regulation, polarity maintenance, cell wall biogenesis/remodeling, septation and septal plugging, signal transduction, transcriptional regulation, vesicle transport, microtubule-based transport and cell cycle regulation) relative to ancestral copy numbers. P-values were calculated by Fisher's exact test, with $P < 0.05$ considered as significant. The percentage of genes retained in yeast genomes was calculated for every functional category by comparing ancient gene copy number prior to the emergence of yeast-like lineages to the average gene copy number of terminals.

**Statistical analyses of gene and domain architectures**. R scripts (Supplementary Data 17) were written to generate coding sequence (CDS)/intron statistics (strand, order, length, count) based on genome annotations of the 71 species. CDS feature coordinates for each gene were extracted and subsequently used to calculate intron coordinates. Statistical significance of the differences between the gene, CDS and intron lengths of 4 unicellular and 39 multicellular fungi was investigated by independent two-tailed Welch's t-test with pooled variance estimation (var.equal = FALSE), using the t.test function in R.

We analyzed changes in domain composition by counting the abundance of domain architectures in plesiomorphically unicellular and multicellular fungi. A domain architecture was defined as the non-overlapping set of PFAM'domain' (not family or clan) signatures detected in a single protein. We first made an inventory of domain architectures in each of the protein families, then compared changes in the frequency of these architectures between proteins of unicellular and multicellular fungi using a generalized linear model ($P < 0.05$) within the same family. We only considered domain architectures that were conserved in at least 70% of the species in that group.

**Reporting summary**. Further information on research design is available in the Nature Research Reporting Summary linked to this article.

## Data availability
The data that support the findings of this study (accession numbers of used genome sequences, copy-number variations for the 72 species, list of hypha morphogenesis genes, list of gene families having duplications in BCZ nodes, and the list of newly identified gene families) are provided as Supplementary Data 1–4 and 8. Results of the statistical tests used in these analyses are found in Supplementary Data 5–9. MCL cluster file of the 72 species is provided as Supplementary Data 10. Gene duplication/loss catalogs, concatenated sequence alignment used for species tree reconstruction, species and gene trees are provided as Supplementary Data 11–16, respectively.

## Code availability
Custom R and Perl code associated with this paper are available as Supplementary Data 17 and at https://github.com/laszlognagy/COMPARE.

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

## Acknowledgements

The authors acknowledge support by the 'Momentum' program of the Hungarian Academy of Sciences (contract No. LP2014/12 to L.G.N.) and the European Research Council (grant no. 758161 to L.G.N.). E.K. acknowledges support from the National Talent Program of the Ministry of Human Capacities (contract No. NTP-NFTÖ-16-0566) and the Young Researchers Program of the Hungarian Academy of Sciences. N.T. acknowledges support from Japan Society for the Promotion of Science (JSPS) KAKENHI (grant number: 18K05545) and Japan Science and Technology Agency (JST) ERATO (grant number: JPMJER1502).

## Author contributions

E.K. and L.G.N. conceived the study. E.K. collected literature data on morphogenesis-related genes. E.K., S.K., L.G.N. and A.N.P. inferred species trees, B.B. performed clustering, E.K., K.K., V.M., T.K., Z.M., and T.V. analyzed gene family evolution. B.H. analyzed gene structural changes. E.K., M.R., N.T. and L.G.N. interpreted the results and wrote the paper. All authors have read and commented on the manuscript.

## Additional information

**Competing interests:** The authors declare no competing interests.

