## [Peer Review File · Nature Communications]

Reviewers' Comments:

Reviewer #1:

Remarks to the Author:

This manuscript by Kiss et al reports that "Comparative genomics reveals the origin of fungal hyphae and multicellularity". The main focus of the study is understanding how genes involved in fungal hyphae are conserved across the fungal kingdom and using those patterns to infer ancestral states. By comparing the genomes of 58 fungi with 13 other eukaryotes, the authors look at the conservation and duplication patterns of genes involved in hyphae, phagocytosis, signaling, and adhesion, as well as search for duplication or loss patterns that correlate with hyphal states. Overall, genes involved in hyphal morphogenesis appear highly conserved, even in fungi with reduced genomes.

Major comments

1. Overall, I miss a clear statement about how this relatively small set of genomes was selected from all the data available across the fungal kingdom to use in these comparative analyses. ie, how well do the selected genomes represent the proportion of hyphal species in the fungal kingdom. There has likely been some selection bias not only in what has been sequenced, but also in the subset of available fungal genome data used for this study.
2. Along these lines, the framing of which species used in this analysis could be considered multicellular was not very clear. While the introduction notes the difficulties in categorizing fungal multicellularity, that then leaves the reader without a clear framework for how to interpret some of the gene conservation results except in the context of species that produce hyphae. Given this and the similar lack of framing of the focus on kinase, receptors and adhesion gene conservation, I question the weight given to noting what this analysis reveals about multicellularity, particularly in the title and abstract.
3. The study of genes involved in hyphal morphogenesis rests on a set of 651 that appear to have been narrowly drawn from two subphyla of ascomycota. What data shows that hyphae in ascomycetes are the same as in the other phyla? The authors should summarize the conservation of these 651 genes, and address the possibility that this data set may limit their power to look at multicellularity across the fungal kingdom. It would also be helpful for the authors to provide a more detailed description of how directly this gene set has been linked a specific role in hyphal morphogenesis.
4. The data availability statement is incomplete; the authors note data sets provided on a 3rd party site that will be provided upon publication—these need to be made available for review. There are also references to a "custom-made Perl script" (L508), "custom Perl script" (L588) and "An R script (available upon request); all these scripts should be made openly available on GitHub or other public repository.

Minor points

L42: "pro-" -> prokaryote.

L117: In Supplementary Table 1, in the row with the genome for *Eremothecium gossypii*, column E notes "Please check their data use policies." Can the authors please confirm if there were restrictions on the use of this data set? Also in this table, the source of all the data should be provided, ie accessions in GenBank. The providers of unpublished data sets used in this work should be acknowledged.

L213-4: For the analysis of CDS and intron lengths, how do the selected categories compare to the properties of all genes? Figure S2 depicts both boxplots and data points, however the data shown is not completely clear-- what is "sum CDS length" and "sum intron length"? Please also add statistical test used to the figure legend.

L228-229: How large was the phagocytotic gene set?

L260-261: How well conserved are these 414 gene families across the species that produce hyphae? In supplementary table 5, many genes appear to be missing in *Aspergillus* but present in yeast

Figure 1- part A appears to have part of the text clipped in overlapping with panel B. Here and elsewhere, the text is quite small and is therefore difficult to read. Also in this figure and elsewhere, the choice to display the gene counts as overlapping circles makes many individual points very difficult to map to specific species.

L134: "backbone" -> backbone

L203-4: Where is the data and significance values reported for the "9 functional groups showed significantly increased numbers of duplications in the cell wall biogenesis and ..."

L346: The smaller number of adhesion related proteins in BCZ nodes may reflect the lower number of such annotated proteins in the less well studied fungi outside of Dikarya and the higher divergence noted in some adhesion related gene families.

L391-394: The noted loss of microtubule systems was an interesting finding; can the authors expand a bit on genes that are lost in yeasts—were these redundant functions, ie duplicated copies, that were lost or any unique genes?

L339-400: Provide a reference for the 'previous reports'.

L508: Provide more detail on how "deep paralogs" were identified with the custom perl script.

L510: What type of "supermatrix" was used for phylogenetic analysis? If sequences were concatenated, this may be more clearly described as a multifasta; otherwise describe how it was converted to a matrix.

L529: How does this MCL protocol differ from that described for the organismal phylogeny section?

L541: What scripts were used from this pipeline? Is this COMPARE software openly available?

Several of the supplementary figures were not visible in the pdf provided for review; they are apparent in the word file. Also supplementary tables 6 and 7 appear to be labeled 5 and 6 in their header rows.

Figure S1- The numbers in the circles are very small; can this text be made any larger? Alternatively, larger circles representing groups of adjacent points (with the range listed) may be another option.

Reviewer #2:

Remarks to the Author:

Title: Comparative genomics reveals the origin of fungal hyphae and multicellularity

Authors: Kiss, Hegedüs, Varga, Merényi, Tamás Kószó, Bálint, Prasanna, Krizsán, Riquelme, Takeshita and Nagy

Background: Major evolutionary transitions (MET) occur when simpler subunits coalesce to form interdependent, self-replicating, semi-autonomous wholes. In their influential monograph, Maynard Smith and Szathmary (1995) identified a number of such transitions, each of which opened the door to quantum leaps in biocomplexity. Many of these also involved transitions in individuality and, therefore, units of selection. Among the most conspicuous of these transitions was the evolution of multicellularity (MC) from unicells (UC), which opened the door to division of labor via cellular differentiation. Further, advent of MC ultimately paved the way to germ line/soma differentiation and the evolution of sex based on gametes rather than on horizontal gene transfer. Multicellularity (MC), unlike eukaryogenesis, has evolved more than two dozen times in the history of life, arising within clades representing all three Domains. This prompted Grosberg and Strathmann (2007) to contend that the evolution of multicellularity is "A minor major transition." Since then, several groups have shown that clonal, as opposed to aggregative, multicellularity can be experimentally evolved in the laboratory, both within the Chlorophyte algae (PMID 30787483) and within the Ascomycetes (PMID: 25600558).

Approach and major findings: Multicellularity has arisen – and been secondarily lost – several times in fungal Kingdom, and when retained, exhibits signatures of convergent evolution (2018 PMID: 29675836). Kiss et al. use a comparative genomics approach to explore this phenomenon, focusing on hyphae, tubular structures that grow by apical extension to form mycelia. Hyphae are a common feature of fungal multicellularity. They can be multinucleated, raising the possibility that genomic conflict can be resolved in different ways than among a group of cells that each have a single nucleus. Also, extant fungal hyphae proliferate via a fractal-like geometry that potentially maximizes assimilation, while minimizing competition. Therefore, Kiss et al. argue that this form of hyphal-based MC may have evolved along different pathway(s) than clonal or aggregative MC. These considerations motivate the authors to reconstruct the genetic origins of hyphae by analyzing the genomes of 4 unicellular fungi, 40 hyphal fungi, 14 secondarily unicellular fungi and 13 non-fungal relatives. They find that hyphal evolution correlates with a multitude of genetic changes, involving hundreds of gene families. Consistent with the unique features of hyphal-based MC, gene families associated with clonal and aggregative MC (e.g., adhesins and sensor kinases) poorly correlate with the advent of hyphae. Instead, what they report are myriad changes in gene structure as well as co-option of ancient eukaryotic genes, including those related to phagocytosis. Indeed, the machinery of hyphal morphogenesis has homologs among non-fungal eukaryotes, as well as true unicellular fungi. Thus, the story of the evolution of hyphae, a trait that defines many fungi, is (or rather should be!) a tale about genetic co-option and evolutionary 'tinkering' sensu Jacob (1977 PMID: 860134).

Major comments: Kiss et al. seek to uncover the origins of a key innovation, hyphae, that opened the door to fungal multicellularity. Their work is a tour de force of phylostratigraphy, as it reveals the complex – and variable – genetic underpinnings of this trait. However, while their approach and their analysis are impeccable and deeply informative, their story-line is weak, undermined by an endless catalogue of gene names and GO terms. Their story lacks a forceful and informed speculation about the particular suite of genetic changes that in different lineages converge on a similar morphology. This is surprising, given that the senior author published in 2018 an interesting article that begins to meditate on this potentially unifying aspect of the present dataset. No reader will question the exhaustive analysis on display here, but every reader will want to be led beyond the facts to what those facts mean. For example, co-option of certain genes early in a given clade's acquisition of hyphae might preclude co-option of others. To be worthy of consideration at Nature Communications

the authors should generate a hypothetical set of alternative rules, or principles, by which they believe this key innovation might have been established. And while it would be beyond the scope of the current project, a thoughtful speculation along those lines could open the door to either in silico testing of those hypotheses, or testing those hypotheses in vivo using synthetic biology.

Minor comments:

The manuscript is generally well-written, if ponderous. While I very much appreciate the authors' scholarship, the MS could benefit from thoughtful editing. Cutting 2 pages from the endless "catalogue" of genes and GO terms could make room for informed speculation about evolutionary patterns within the 8-11 clades that gained and retained MC and the clades that lost it secondarily. Also, most readers would like to hear about the authors opinions on whether the "invention" of hyphae opened the door to cellular differentiation in higher fungi, and if not, what did.

II. 206. Are the authors suggesting the fungi as a single clade underwent in concert a "period of extensive gene duplication"? Is there compelling evidence for such an assertion?

II. 222 and throughout (e.g., II. 471 and II. 481) "significant changes in gene structure" The level of analysis of "gene structure" in this work is rudimentary, being chiefly focused on CDS and intron length rather than on functional domains, and predicted higher-order structures. I recognize that these deeper analyses are computationally intense. However, the authors should not overstate the significance of their "gene structure" findings, as one could imagine many, many ways that biologically significant changes in structure could not be discovered by their analyses.

II. 249. and II. 258. On philosophical grounds I object to the use of the term "evolutionary dynamics" in Fig. 3. Strictly speaking, evolutionary dynamics is the unfolding of mathematical principles governing how populations of biological entities evolve, and falls within the domain of population genetics. This figure and the accompanying discussion are gene genealogies in the absence of population data, even that of all potential variants extant at each node.

II. 480 And just what might be those "multiple mechanisms"? This is the essence of my chief concern about this paper, and my conviction that it must go beyond phenomenology to informed speculation about different genetic pathways to hyphal MC.

II. 486 Just what is that "peculiar life history"?

Supplementary Tables 5 and 6 are mislabeled (see upper left heading on each file). One Supplementary Table appears to be absent.

Reviewer #3:

Remarks to the Author:

Review of "Comparative genomics reveals the origin of fungal hyphae and multicellularity"

The authors perform comparative genomic analyses to infer the genetic changes involved in the evolution of hyphae. This is an important question and, in general, the methodology is appropriate, the results are interesting, and the interpretation is good.

I have, however, some major concerns with regards to the manuscript.

Major comments

My most important concern is with regards the original phylogeny on which the comparative genomic analyses are based.

There are two problems with this phylogeny. The first one is methodological, the second implies missing taxa. The tree has been obtained using only maximum likelihood and with a very simple model of evolution. Given the importance of this tree, I think the authors should redo the tree with a better model of evolution (or justify why the use of PROTGAMMAWAG). Moreover, the authors have not included *Fonticula alba*, the closest non-fungal outgroup. I think they should, being such an important taxa. It may as well be that a more complex model of evolution and the addition of *F. alba* does not change at all the results, but I will feel more confident if those changes are made.

Moreover, some figures are a bit difficult to read and interpret. The authors may consider doing them larger or change them to make them more clear.

Minor comments

Page 1, line 28 "co-option..phagocytosis-related genes" Be careful, it is not clear phagocytosis genes detected are indeed only for phagocytosis.

Page 1, line 36 "contrary to most multicellular lineages..." Please tone down. We know very little about "most multicellular lineages"

Page 1, line 38 "fungi took a unique route to multicellularity that involved gene family diversification and extensive co-option of ancestral genes". It does not seem so "unique", animals seem to have done something similar!

Page 2, line 59 "syncytial body (e.g. *Capsaspora*). " *Capsaspora* does not go through a syncytial body, but rather forms multicellular structures by cell aggregation. *Ichthyosporeans* do go through a syncytial (better named as coenocyte) body.

Page 4, line 134. "beckbone". Change to "backbone".

Page 6, line 193. "their ancestors". What do the authors mean?. Please clarify.

Page 15, line 459. I think Suga et al. should be cited here. Nature Communications 4: 2325 doi: 10.1038/ncomms3325 (2013).

Figure 1. Nodal support (bootstrap, for example) should be indicated

Figure 2. This figure and the text is confusing Genes or gene families? It is not clear in the text and the figure. Usually, one will expect that gene families will appear before, and then those gene families are further expanded in more complex (multicellular) lineages. Is that what the authors see? It is not clear...

Reviewer #1:

This manuscript by Kiss et al reports that “Comparative genomics reveals the origin of fungal hyphae and multicellularity”. The main focus of the study is understanding how genes involved in fungal hyphae are conserved across the fungal kingdom and using those patterns to infer ancestral states. By comparing the genomes of 58 fungi with 13 other eukaryotes, the authors look at the conservation and duplication patterns of genes involved in hyphae, phagocytosis, signaling, and adhesion, as well as search for duplication or loss patterns that correlate with hyphal states. Overall, genes involved in hyphal morphogenesis appear highly conserved, even in fungi with reduced genomes.

Major comments

Reviewer: 1. Overall, I miss a clear statement about how this relatively small set of genomes was selected from all the data available across the fungal kingdom to use in these comparative analyses. ie, how well do the selected genomes represent the proportion of hyphal species in the fungal kingdom. There has likely been some selection bias not only in what has been sequenced, but also in the subset of available fungal genome data used for this study.

Authors: We have included a justification of taxon sampling in the Methods section. We selected taxa so as to represent each of the major fungal clades, including plesiomorphically unicellular, filamentous and secondarily unicellular species. We included all clades that contain filamentous species, so we are confident that the dataset appropriately represents the diversity of morphological types in the fungal kingdom. No attempt was made to include all publicly available genome in the dataset, as that would have yielded a dataset of a computationally intractable size.

Reviewer: 2. Along these lines, the framing of which species used in this analysis could be considered multicellular was not very clear. While the introduction notes the difficulties in categorizing fungal multicellularity, that then leaves the reader without a clear framework for how to interpret some of the gene conservation results except in the context of species that produce hyphae. Given this and the similar lack of framing of the focus on kinase, receptors and adhesion gene conservation, I question the weight given to noting what this analysis reveals about multicellularity, particularly in the title and abstract.

Authors: Cellularity level of each of the species is presented on Figure 1 (dots next to species names). In the revised submission, we included this information in Supplementary Table 1 also. Our focus is on hyphal growth, which represents a unique type of multicellular growth. Other multicellular lineages were included in the dataset with the purpose of comparing patterns of gene family dynamics in multicellular fungi to that of better characterized multicellular lineages (e.g. animals, Dictyostelium). We do not aim to make a comprehensive comparison of independent multicellular lineages. In the revised ms we moved the section on kinase, receptor and adhesive repertoires earlier in the text and improved the storyline to highlight the reason for focusing on these groups of genes.

Reviewer: 3. The study of genes involved in hyphal morphogenesis rests on a set of 651 that appear to have been narrowly drawn from two subphyla of ascomycota. What data shows that hyphae in ascomycetes are the same as in the other phyla?

Authors: There are several studies that show that mechanisms of hyphal morphogenesis are shared by all or most hyphal fungi (see e.g. Roberson et al, 2010, in Cellular and Molecular Biology of Filamentous Fungi; Harris 2011, in Fungal Biology 115 (6); Wessels, 1986, International Review of Cytology, 104:37-79). For example, Schizophyllum (Basidiomycota) uses the same general mechanisms for hyphal morphogenesis as does Aspergillus, Neurospora or hypha-forming Candida species. Accordingly, most of the 362 gene families (which comprise the 651 genes) were shared by all or most filamentous fungi, indicating that our analyses are not restricted to specific clades, but address the general mechanisms of hypha morphogenesis. Species-specific morphogenetic genes exist and have been included in our analyses for completeness, but these do not interfere with our conclusions on the early events of fungal evolution.

Reviewer: The authors should summarize the conservation of these 651 genes, and address the possibility that this data set may limit their power to look at multicellularity across the fungal kingdom. It would also be helpful for the authors to provide a more detailed description of how directly this gene set has been linked a specific role in hyphal morphogenesis.

Authors: Each gene has been linked to hyphal morphogenesis by detailed mechanistic studies. These references, the role of each gene in hypha morphogenesis, as well as its source, homologues, and cluster ID can be found in Supplementary Table 3. Length restrictions preclude a more detailed description of these genes in the main text (see also note by Reviewer 2 to shorten descriptions of fungal gene families). When assembling the dataset of 651 genes, we mined relevant papers until no new genes could be found. This ensures that all or the vast majority of known hypha morphogenesis genes are included in this study. Because hyphal morphogenesis is a well-known process, this also suggests that much of its genetic toolkit is included in our study. Further, genes with a hitherto unknown role in hyphal morphogenesis certainly exist, but we expect these to be the minority and their identification is beyond the scope of this paper.

To summarize the conservation of the 651 genes, we constructed a presence/absence plot of the 362 gene families that contain these genes. This is presented as Supplementary Figure 4.

Reviewer: 4. The data availability statement is incomplete; the authors note data sets provided on a 3rd party site that will be provided upon publication—these need to be made available for review. There are also references to a “custom-made Perl script” (L508), “custom Perl script” (L588) and “An R script (available upon request); all these scripts should be made openly available on GitHub or other public repository.

Authors: We have submitted all datasets related to this paper as supplementary files and material in the ms submission system, including novel scripts that were used in the analyses. Scripts of the COMPARE pipeline are available on GitHub, for which a link is now provided in the Methods.

Minor points

Reviewer: L42: “pro-“ -> prokaryote.

Authors: Corrected.

Reviewer: L117: In Supplementary Table 1, in the row with the genome for *Eremothecium gossypii*, column E notes “Please check their data use policies.” Can the authors please confirm if there were restrictions on the use of this data set? Also in this table, the source of all the data should be provided, ie accessions in GenBank. The providers of unpublished data sets used in this work should be acknowledged.

Authors: We confirm that all genomic data used in this work are published (for Eremothecium see e.g. Dietrich FS, et al. The Ashbya gossypii genome as a tool for mapping the ancient Saccharomyces cerevisiae genome. Science. 2004 Apr 9;304(5668):304-7). We have updated this table with the correct reference to the publication of each species' genome.

Reviewer: L213-4: For the analysis of CDS and intron lengths, how do the selected categories compare to the properties of all genes? Figure S2 depicts both boxplots and data points, however the data shown is not completely clear-- what is “sum CDS length” and “sum intron length”? Please also add statistical test used to the figure legend.

Authors: This analysis is restricted to hyphal morphogenesis genes, as stated in the text and the legend and compared CDS and intron lengths of plesiomorphically unicellular and filamentous fungi. We added the name of the statistical test to the legend. The terms “sum CDS length” and “sum intron length” were misleading, we changed them in the figure. They refer to mean coding sequence length and intron lengths, respectively.

Reviewer: L228-229: How large was the phagocytotic gene set?

Authors: It contained 106 genes. We added this to the text.

Reviewer: L260-261: How well conserved are these 414 gene families across the species that produce hyphae? In supplementary table 5, many genes appear to be missing in *Aspergillus* but present in yeast

Authors: On average, each of these genes were present in >70% of filamentous species. Indeed, some (20 of the 414) genes are missing from Aspergillus, but in our opinion this does not undermine their potential role in hyphal growth, because different lineages might be using somewhat different gene sets for hyphal growth. To show the conservation of the 414 gene across the dataset, we included a heatmap of gene copy numbers in Figure 5.

Reviewer: Figure 1- part A appears to have part of the text clipped in overlapping with panel B. Here and elsewhere, the text is quite small and is therefore difficult to read. Also in this figure and elsewhere, the choice to display the gene counts as overlapping circles makes many individual points very difficult to map to specific species.

Authors: We revised Figure 1, for us all parts of the figure are visible on the current version (but we note that inserting the figures into the text file compresses them more than they will be compressed online or in a pdf).

We set a paler gray color and a solid black line for the individual circles on panel C to make them more visible. The message these circles should convey is the distribution of copy numbers across the tree, which, we think the figure shows appropriately. We revised the rest of the figures to make mapping of circles to species easier.

Reviewer: L134: "backbone" -> backbone

Authors: Corrected.

Reviewer: L203-4: Where is the data and significance values reported for the "9 functional groups showed significantly increased numbers of duplications in the cell wall biogenesis and ..."

Authors: We included p-values in panel C of Figure 1.

Reviewer: L346: The smaller number of adhesion related proteins in BCZ nodes may reflect the lower number of such annotated proteins in the less well studied fungi outside of Dikarya and the higher divergence noted in some adhesion related gene families.

Authors: We agree that only a few adhesive proteins are known in non-dikarya. Nevertheless, because gene families that expanded in the Dikarya were present in ancestral fungi, they had the possibility for an expansion if selection dictated so. We added a sentence on this in the revised ms.

Reviewer: L391-394: The noted loss of microtubule systems was an interesting finding; can the authors expand a bit on genes that are lost in yeasts—were these redundant functions, ie duplicated copies, that were lost or any unique genes?

Authors: These were mostly non-redundant functions and losses erased some gene families completely from yeast genomes. We rephrased this sentence to be more specific on what functions are lost completely and which are only contracted. Information on losses is also shown on Fig. 6/b.

Reviewer: L339-400: Provide a reference for the 'previous reports'.

Authors: We rephrased this sentence and added a reference to previous works.

Reviewer: L508: Provide more detail on how "deep paralogs" were identified with the custom perl script.

Authors: We rephrased this section and added a reference to the complete description of the algorithm (Prasanna et al Systematic Biology in press).

Reviewer: L510: What type of “supermatrix” was used for phylogenetic analysis? If sequences were concatenated, this may be more clearly described as a multifasta; otherwise describe how it was converted to a matrix.

Authors: The phrase supermatrix here refers to the concatenated single-gene alignments, in the sense this phrase is often used in phylogenetics (see de Queiroz and Gatesy 2007 Trends in Ecol &Evol.). The term multifasta would suggest sequences were concatenated prior to alignment. This is not the case here, we first inferred single-gene alignments, which were then concatenated. This strategy effectively retained gene boundaries across gene families and allowed us to use partitioned models in phylogenetic analyses.

Reviewer: L529: How does this MCL protocol differ from that described for the organismal phylogeny section?

Authors: The two protocols referred to the same analysis. We removed the ambiguity from the text.

Reviewer: L541: What scripts were used from this pipeline? Is this COMPARE software openly available?

Authors: Yes. We added a link to the Github repository.

Reviewer: Several of the supplementary figures were not visible in the pdf provided for review; they are apparent in the word file. Also supplementary tables 6 and 7 appear to be labeled 5 and 6 in their header rows.

Authors: We double checked the correct insertion of the Supplementary figures in the revised ms package. We corrected the numbering of Supplementary Table 6 and 7.

Reviewer: Figure S1- The numbers in the circles are very small; can this text be made any larger? Alternatively, larger circles representing groups of adjacent points (with the range listed) may be another option.

Authors: We enlarged both the numbers in circles (now moved out of the circles) and the list of gene names below the figure.

Reviewer #2 (Remarks to the Author):

Reviewer:

Major comments: Kiss et al. seek to uncover the origins of a key innovation, hyphae, that opened the door to fungal multicellularity. Their work is a tour de force of phylostratigraphy, as it reveals the complex – and variable – genetic underpinnings of this trait. However, while their approach and their analysis are impeccable and deeply informative, their story-line is weak, undermined by an endless catalogue of gene names and GO terms. Their story lacks a forceful and informed speculation about the particular suite of genetic changes that in different lineages converge on a similar morphology. This is surprising, given that the senior author published in 2018 an interesting article that begins to meditate on this potentially

unifying aspect of the present dataset. No reader will question the exhaustive analysis on display here, but every reader will want to be led beyond the facts to what those facts mean. For example, co-option of certain genes early in a given clade's acquisition of hyphae might preclude co-option of others. To be worthy of consideration at Nature Communications the authors should generate a hypothetical set of alternative rules, or principles, by which they believe this key innovation might have been established. And while it would be beyond the scope of the current project, a thoughtful speculation along those lines could open the door to either in silico testing of those hypotheses, or testing those hypotheses in vivo using synthetic biology.

Authors: We appreciate this suggestion. We have improved the storyline of the paper by emphasizing working hypotheses and by rearranging the sections of the paper to reflect our line of thoughts better. We have also included a discussion on what prerequisites hyphal growth might have had and how its evolution contributed to further increases in complexity in fungi. We speculate that the particular set of changes and innovations eventually locked fungi into converging on hyphae as the single most optimal solution for multicellular growth. This might have been caused by several individual tweaks to growth mode, above all, the loss of phagocytosis and pseudopodia, the emergence of the cell wall and osmotrophy.

We appreciate referencing our 2018 review, but that paper focused on complex multicellularity which is, indeed, convergent in fungi. In this paper we focus on hyphal multicellularity, which is different from complex multicellularity and shows a single origin in fungi. This is the reason we do not discuss these data in a convergence context.

Minor comments:

Reviewer: The manuscript is generally well-written, if ponderous. While I very much appreciate the authors' scholarship, the MS could benefit from thoughtful editing. Cutting 2 pages from the endless "catalogue" of genes and GO terms could make room for informed speculation about evolutionary patterns within the 8-11 clades that gained and retained MC and the clades that lost it secondarily. Also, most readers would like to hear about the authors opinions on whether the "invention" of hyphae opened the door to cellular differentiation in higher fungi, and if not, what did.

Authors: We have shortened the listing of fungal gene families and GO terms and improved the storyline of the ms. We inserted a discussion of how hyphae might have contributed to further morphological diversification in fungi. We have done our best to find a good balance between general considerations of the evolution of multicellularity and mycological conclusions and information (e.g. gene families). Given the size of the community working on hypha morphogenesis in mycology, we envision a significant interest from fungal geneticists in this work. Therefore, we aim for a proper balance between the above two aspects and we think the ms has improved significantly in this aspect.

Reviewer: II. 206. Are the authors suggesting the fungi as a single clade underwent in concert a "period of extensive gene duplication"? Is there compelling evidence for such an assertion?

Authors: No, we find no evidence for that. This sentence reflects to predictions that a major transition like multicellularity should coincide with major genetic innovations (e.g. Knoll 2011), for which we see no evidence here. We inserted a reference in the sentence to make this clear.

Reviewer: ll. 222 and throughout (e.g., ll. 471 and ll. 481) “significant changes in gene structure” The level of analysis of “gene structure” in this work is rudimentary, being chiefly focused on CDS and intron length rather than on functional domains, and predicted higher-order structures. I recognize that these deeper analyses are computationally intense. However, the authors should not overstate the significance of their “gene structure” findings, as one could imagine many, many ways that biologically significant changes in structure could not be discovered by their analyses.

Authors: We have toned down these claims and performed a new analysis of domain evolution in relation to the evolution of hyphae.

Reviewer: ll. 249. and ll. 258. On philosophical grounds I object to the use of the term “evolutionary dynamics” in Fig. 3. Strictly speaking, evolutionary dynamics is the unfolding of mathematical principles governing how populations of biological entities evolve, and falls within the domain of population genetics. This figure and the accompanying discussion are gene genealogies in the absence of population data, even that of all potential variants extant at each node.

Authors: We have changed 'evolutionary dynamics' to 'evolutionary history'.

Reviewer: ll. 480 And just what might be those “multiple mechanisms”? This is the essence of my chief concern about this paper, and my conviction that it must go beyond phenomenology to informed speculation about different genetic pathways to hyphal MC.

Authors: We have deleted this sentence and included discussion elsewhere in the ms on how these mechanisms might have contributed to the evolution of hyphae.

Reviewer: ll. 486 Just what is that “peculiar life history”?

Authors: We refer to the unique growth mode and organization of hyphae, rephrased this sentence.

Reviewer: Supplementary Tables 5 and 6 are mislabeled (see upper left heading on each file). One Supplementary Table appears to be absent.

Authors: We corrected the supplementary tables.

Reviewer #3 (Remarks to the Author):

Reviewer: My most important concern is with regards the original phylogeny on which the comparative genomic analyses are based. There are two problems with this phylogeny. The first one is methodological, the second implies missing taxa. The tree has been obtained using only maximum likelihood and with a very simple model of evolution. Given the

importance of this tree, I think the authors should redo the tree with a better model of evolution (or justify why the use of PROTGAMMAWAG). Moreover, the authors have not included *Fonticula alba*, the closest non-fungal outgroup. I think they should, being such an important taxa. It may as well be that a more complex model of evolution and the addition of *F. alba* does not change at all the results, but I will feel more confident if those changes are made.

Authors: We added Fonticula to the dataset and performed new phylogenomic analyses with the new dataset (at the time of dataset assembly its genome was not public, and it still is in the 'gray' zone of genomes with not officially having been published, this is why we did not include it initially). We inferred new trees under three models, including partitioned stationary models (RAxML), the CAT model (in Phylobayes) which handles heterogeneity in the data and the evolutionary process, as well as partitioned free-rate models (IQ-Tree). We consider these models as some of the most sophisticated ones currently available for genome-scale datasets. The new trees are completely congruent with the old tree and Fonticula was placed as the sister group to Fungi, as expected based on previous studies. All three trees are shown as Supplementary Figure 1 and support values were added to Figure 1 in the main text.

We re-run the ancestral character state reconstructions on the new trees that include Fonticula. This did not change the results, but the new tree and results are included in Figure 1.

We also re-run comparative genomic analyses with Fonticula added to the dataset: we obtained a new clustering with this species and examined the copy number distribution of hyphal morphogenesis. These analyses showed that Fonticula's gene repertoire is similar to those of non-fungal protists (see Supplementary Note 1, which was created to summarize Fonticula's multicellularity-related gene repertoire), which led us to conclude that it does not influence our conclusions on hyphal multicellularity in fungi. Because of this and the excessive computational run times of the COMPARE analyses, we retained the original COMPARE analyses in the manuscript, but we supplemented each figure with the copy-number data of Fonticula.

Reviewer: Moreover, some figures are a bit difficult to read and interpret. The authors may consider doing them larger or change them to make them more clear.

Authors: We have revised the figures, increased font size wherever it was possible.

Minor comments

Reviewer: Page 1, line 28 “co-option..phagocytosis-related genes” Be careful, it is not clear phagocytosis genes detected are indeed only for phagocytosis.

Authors: We rephrased this sentence slightly. Nevertheless, of the several interpretations of phagocytosis machinery put forth recently, our analysis followed a narrowly defined set of genes, as described in Bozzaro et al 2008, Riveri 2008 and Mao and Finneman 2015.

Reviewer: Page 1, line 36 “contrary to most multicellular lineages...” Please tone down. We know very little about “most multicellular lineages”

Authors: We toned down this statement.

Reviewer: Page 1, line 38 “fungi took a unique route to multicellularity that involved gene family diversification and extensive co-option of ancestral genes”. It does not seem so “unique”, animals seem to have done something similar!

Authors: We agree. “Unique” referred to the multicellular organization and its emergence through putatively rhizoid-like intermediates in fungi. We rephrased the sentence.

Reviewer: Page 2, line 59 “syncitial body (e.g. Capsaspora). “ Capsaspora does not go through a syncitial body, but rather forms multicellular structures by cell aggregation. Ichthyosporeans do go through a syncitial (better named as coenocyte) body.

Authors: We replaced Capsaspora with a reference to Ichthyosporeans.

Reviewer: Page 4, line 134. “beckbone”. Change to “backbone”.

Authors: Corrected.

Reviewer: Page 6, line 193. “their ancestors”. What do the authors mean?. Please clarify.

Authors: We referred to ancestral nodes predating fungi that are shared with other eukaryotes. We rephrased the sentence.

Reviewer: Page 15, line 459. I think Suga et al. should be cited here. Nature Communications 4: 2325 doi: 10.1038/ncomms3325 (2013).

Authors: We inserted the citation.

Reviewer: Figure 1. Nodal support (bootstrap, for example) should be indicated

Authors: We added nodal support from the three types of phylogenetic analyses.

Reviewer: Figure 2. This figure and the text is confusing Genes or gene families? It is not clear in the text and the figure. Usually, one will expect that gene families will appear before, and then those gene families are further expanded in more complex (multicellular) lineages. Is that what the authors see? It is not clear...

Authors: We rephrased the legend of this figure and in the text to clarify that we refer to gene families, which are represented by the experimentally characterized Aspergillus orthologue on the figure.

Reviewers' Comments:

Reviewer #1:

Remarks to the Author:

The authors have addressed all of the comments in my review. Regarding the addition of the genome of *Fonticula alba*, it would be courteous to acknowledge the use of all the unpublished genomes - *Sphaeroforma arctica* and *Batrachochytrium dendrobatidis* as well as it appears from Table S1, ie the UNICORN project (<https://www.ncbi.nlm.nih.gov/pubmed/?term=17275133>), the Broad Institute, and JGI and their user community.

Reviewer #2:

Remarks to the Author:

I have reviewed the revision response letter for and revision of "Comparative genomics reveals the origin of fungal hyphae and multicellularity" by Kiss et al. The manuscript is markedly improved relative to the initial submission; in particular, the Results and Discussion/Conclusions now flow much more smoothly, and follow a stronger thematic trajectory. Still, the Introduction and early Results sections suffer from grammatical/rhetorical errors and unclear messaging. I have taken the liberty of offering corrections, suggested changes and a few remaining questions on the manuscript itself, attached below. Also, (and I mean this no disrespect to the authors) may I suggest that prior to submitting another revision they take advantage of Nature's editing service for a last read : <https://authorservices.springernature.com/language-editing/>

Thank you for giving me the opportunity to review this interesting work.

Reviewer #3:

Remarks to the Author:

thanks the authors for the revision

Answers to Reviewers' comments.

Reviewer #1 (Remarks to the Author):

The authors have addressed all of the comments in my review. Regarding the addition of the genome of *Fonticula alba*, it would be courteous to acknowledge the use of all the unpublished genomes - *Sphaeroforma arctica* and *Batrachochytrium dendrobatidis* as well it appears from Table S1, ie the UNICORN project (<https://www.ncbi.nlm.nih.gov/pubmed/?term=17275133>), the Broad Institute, and JGI and their user community.

We contacted lead PIs of these genomes and obtained permissions for their use.

Reviewer #2 (Remarks to the Author):

I have reviewed the revision response letter for and revision of "Comparative genomics reveals the origin of fungal hyphae and multicellularity" by Kiss et al. The manuscript is markedly improved relative to the initial submission; in particular, the Results and Discussion/Conclusions now flow much more smoothly, and follow a stronger thematic trajectory. Still, the Introduction and early Results sections suffer from grammatical/rhetorical errors and unclear messaging. I have taken the liberty of offering corrections, suggested changes and a few remaining questions on the manuscript itself, attached below. Also, (and I mean this no disrespect to the authors) may I suggest that prior to submitting another revision they take advantage of Nature's editing service for a last read :

<https://authorservices.springernature.com/language-editing/>

Thank you for giving me the opportunity to review this interesting work.

We appreciate the edits and the suggestions. We incorporated these in the ms and have double-checked the text for linguistic issues.

Reviewer #3 (Remarks to the Author):

thanks the authors for the revision